# Hierarchically Porous Carbon Networks Derived from Chitosan for High-Performance Electrochemical Double-Layer Capacitors

**DOI:** 10.3390/nano13222961

**Published:** 2023-11-16

**Authors:** Kwang Hyun Park, Segi Byun, Boemjin Ko, Woong-Gil Hong, Jungmo Kim, Dongju Lee, Wang Geun Shim, Sung Ho Song

**Affiliations:** 1Division of Advanced Materials Engineering, Center for Advanced Materials and Parts of Powders, Kongju National University, Cheonan-si 31080, Republic of Korea; recite14@kongju.ac.kr (K.H.P.); qjawls5004@smail.kongju.ac.kr (B.K.); 2Hydrogen Convergence Materials Laboratory, Korea Institute of Energy Research (KIER), 152 Gajeong-ro, Yuseong-gu, Daejeon 34129, Republic of Korea; segibyun@kier.re.kr; 3Energy Engineering, University of Science and Technology (UST), Daejeon 34113, Republic of Korea; 4Department of Chemical Engineering, Sunchon National University, 255 Jungang-ro, Suncheon-si 57922, Republic of Korea; hwg7524@scnu.ac.kr; 5Nano Hybrid Technology Research Center, Korea Electrotechnology Research Institute (KERI), Changwon 51543, Republic of Korea; jungmokim@keri.re.kr; 6Department of Urban, Energy, and Environmental Engineering, Chungbuk National University, Chungdae-ro 1, Seowon-gu, Cheongju 28644, Republic of Korea

**Keywords:** activated carbon, biomass, electrochemical capacitor, ultra-micropores, chitosan

## Abstract

Activated carbon (AC) compounds derived from biomass precursors have garnered significant attention as electrode materials in electric double-layer capacitors (EDLCs) due to their ready availability, cost-effectiveness, and potential for mass production. However, the accessibility of their active sites in electrochemistry has not been investigated in detail. In this study, we synthesized two novel macro/micro-porous carbon structures prepared from a chitosan precursor using an acid/potassium hydroxide activation process and then examined the relationship between their textural characteristics and capacitance as EDLCs. The material characterizations showed that the ACs, prepared through different activation processes, differed in porosity, with distinctive variations in particle shape. The sample activated at 800 °C (*Act*-chitosan) was characterized by plate-shaped particles, a specific surface area of 4128 m^2^/g, and a pore volume of 1.87 cm^3^/g. Assessment of the electrochemical characteristics of *Act*-chitosan showed its remarkable capacitance of 183.5 F/g at a scan rate of 5 mV/s, and it maintained exceptional cyclic stability even after 10,000 cycles. The improved electrochemical performance of both chitosan-derived carbon structures could thus be attributed to their large, well-developed active sites within pores < 2 nm, despite the fact that interconnected macro-porous particles can enhance ion accessibility on electrodes. Our findings provide a basis for the fabrication of biomass-based materials with promising applications in electrochemical energy storage systems.

## 1. Introduction

With the rapid increase in demand for energy storage devices with high energy/power density, electrochemical capacitors have emerged as promising candidates for manufacturing operations, emergency power supplies, military applications, electric vehicles, and other applications [1]. Among the electrical energy storage systems of current interest are electric double-layer capacitors (EDLCs), which store electrical energy by forming a double layer of electrolyte ions on electrode surfaces. EDLCs offer significant advantages, including high power density, excellent rate capability, and stable capacitance during long-term cycling, without decay or fading, due to their reversible surface-dominant adsorption/desorption mechanism. These electrochemical characteristics are generally influenced by the textural properties of the active electrode, such as the specific surface area (SSA), pore size, pore size distribution (PSD), and oxygen functionality [2,3,4].

While the electrochemical performance of carbonaceous materials, including carbon nanotubes (CNTs), graphene, aerogels, and fibers, continues to improve [5,6], the fabrication processes for these materials are still plagued by high production costs and their inherent complexity. Biomass-based carbons hold promise as electrode materials [7,8,9] given advantages such as their abundance, non-toxic nature, and cost-effectiveness. However, their relatively low SSA and non-uniform pore distribution have hindered their performance as EDLCs. To enhance the capacitive performance of biomass-based carbons, materials with well-defined porosity within optimized pore ranges and increased SSA are required. These features are essential to the efficient migration of ions between electrolytes and the electrode surface. One promising approach involves introducing heteroatoms, such as B, N, and S, into the carbon framework using various techniques to enhance the capacitive behavior of the materials through faradaic charge-transfer reactions. Considerable research has been devoted to obtaining porous carbon materials with a high SSA and introducing heteroatoms to improve their specific capacitance. For instance, activated carbons (ACs) derived from biomaterials such as wood have been extensively studied. Efforts aimed at addressing the challenges associated with porosity and surface area, and at harnessing the potential of heteroatoms, are paving the way for the development of high-performance EDLCs using biomass-based carbons, such as [10,11], peat [12], coconut [13], and others [14].

Chitosan, a biopolysaccharide primarily composed of 2-amino-2-deoxy-D-glucose units, is the second most abundant biopolymer on Earth after cellulose. Due to its nitrogen-rich composition and distinct acid-base properties, chitosan has garnered significant attention as a carbon precursor, with numerous studies focusing on its use in the preparation of carbon materials [15,16,17,18,19]. Kucinska et al. developed a practical approach to convert chitosan into AC by utilizing a Na_2_CO_3_ solution as an activator agent, resulting in a porous carbon material with a relatively low surface area of 400 m^2^/g [20]. Another common method of chitosan activation involves dissolving it in dilute acetic acid followed by hydrothermal carbonization at a mild temperature [21]. The synthesis of ACs derived from chitosan through a chemical activation process using potassium hydroxide (KOH) as the activation agent has also been explored [22,23]. There have been several successful demonstrations of the fabrication of N-doped carbon materials from chitosan, whose amide and amino groups enable the incorporation of nitrogen functionalities into the carbon framework [24,25,26]. Recently, a novel approach involving the hydrothermal carbonization of chitosan with the addition of acetic acid, followed by a KOH-based chemical activation process, was described [18,19]. These methods have yielded hierarchical porous carbons with distinctive honeycomb-like frameworks and Brunauer–Emmett–Teller (BET) surface areas >3500 m^2^/g. The resulting carbon material demonstrated favorable electrochemical properties despite the relatively low N-doping level (~1 at%). Moreover, the presence of a limited number of active sites, attributed to randomly developed pores, irregular particle shapes, and low production yields, presents persistent challenges. Therefore, it is essential to establish a scalable synthesis method for active materials with uniformly distributed nanopores measuring less than 2 nm. This synthesis approach is pivotal for achieving enhanced performance metrics, including increased capacitance, retention of capacitance at higher scan rates, and consistent capacitance throughout extended cycles in energy storage systems.

In this study, we introduce a straightforward and accessible process for the activation of chitosan-derived carbon in the creation of highly porous carbon structures with a large SSA, a high pore volume, and different types of particle shapes. The process involves carbonization of the biomass-derived chitosan precursor followed by acid or KOH activation. Porous carbons were prepared from acid- or KOH-activated carbonized chitosan, and their electrochemical properties were then evaluated when used as active electrodes for EDLCs. A correlation between available active sites and capacitance was determined in a comprehensive analysis utilizing N_2_ adsorption/desorption, X-ray photoelectron spectroscopy, and cyclic voltammetry (CV) at varying scan rates. The impressive SSA (4128 m^2^/g) and pore volume (1.87 cm^3^/g) of *Act*-chitosan accounted for its outstanding capacitance, which reached 183.5 F/g at a scan rate of 5 mV/s, while maintaining a remarkable cyclic stability even after 10,000 cycles. The enhanced electrochemical performance of *Act*-chitosan resulting from its high SSA and high pore volume, particularly within optimized pores < 2 nm in size, remains significant despite the limited development of porous carbon networked particles able to facilitate ion migration and electron pathways. The successful fabrication of high-performance ACs from biomass-derived chitosan, exemplified by *Act*-chitosan and HNO_3_-*Act*-chitosan, opens up exciting possibilities for the development of novel biomass-based materials with applications in electrochemical energy storage systems.

## 2. Materials and Methods

### 2.1. Fabrication of Porous Activated Carbons from Chitosan Precursor

The chitosan utilized in this study (molecular mass:190,000–300,000 Da) was purchased from Sigma-Aldrich (St Louis, MO, USA). The chitosan was dried in a vacuum oven at 60 °C and then carbonized for 2 h at 400 °C. The pyrolyzed chitosan was subsequently ground into a fine powder and dried for 24 h at 60 °C. Active material with a porous structure was fabricated by subjecting one of the powdered samples to a 1-h acid treatment using 1 M HNO_3_. The chemically treated sample and an unaltered powdered sample were combined with KOH in a mass ratio of 1:4. These mixtures were activated in an argon (Ar) gas atmosphere for 1 h at 800 °C, thoroughly rinsed with distilled hot water to neutralize the pH, and dried in an oven at 100 °C. The resulting activated porous samples are referred to herein as HNO_3_-*Act*-chitosan and *Act*-chitosan.

### 2.2. Characterizations

The morphology of the samples was analyzed using field-emission scanning electron microscopy (FE-SEM; XL30SFEG; Philips, Amsterdam, The Netherlands). High-resolution transmission electron microscopy (HR-TEM; Tecnai G2 F30; FEI, Hillsboro, OR, USA) was performed, after grinding the powdered samples, by placing a droplet of the sample dispersed in EtOH on the TEM grid. The chemical components of the samples were determined using X-ray photoelectron spectroscopy (XPS; Sigma Probe; Thermo, Waltham, MA, USA). Raman spectroscopy (excitation at 514 nm) using an N8 NEOS and Raman AFM/SPM system (Senterra; Bruker GmbH, Bremen, Germany) and X-ray diffraction (XRD; Rigaku, Tokyo, Japan) were also performed. The textural properties of the samples were analyzed by N_2_ adsorption/desorption measurements at 77 K (Tristar II volumetric adsorption analyzer; Micromeritics, Norcross, GA, USA). The SSA, pore volume, and pore width were calculated by applying the BET and two-dimensional-non-local density functional theory (2D-NLDFT) adsorption models. The surface heterogeneity of the samples was investigated by applying the nitrogen adsorption energy distribution (AED) function [27,28].

### 2.3. Electrochemical Characterization

The electrode slurries were prepared by combining 80 wt% of the chitosan-based samples (*Carb*-chitosan, HNO_3_-*Act*-chitosan, *Act*-chitosan), 10 wt% carbon black (Super-P), and 10 wt% binder (polyvinylidene fluoride [PVDF]; Solef6020). A Ni current collector was coated with the slurries using a doctor blade coater and dried for 12 h at 100 °C in a vacuum oven. The electrode was stamped with an active area of 1 cm^2^ and a mass loading of 2–3 mg/cm^2^ and then assembled in a two-electrode cell configuration (ECC-Aqu, EL-Cell), with 6 M KOH as the electrolyte. All electrochemical analyses were conducted under ambient conditions using a potentiostat (SP-150; Bio-Logic Science Instruments, Seyssinet-Pariset, France). Electrochemical impedance spectroscopy (EIS) measurements were obtained within a frequency range of 100 kHz to 10 mHz, applying a sinusoidal wave of 10 mV at open circuit voltage. Additionally, CV curves were generated at scan rates ranging from 5 to 500 mV/s, with a potential window ranging from 0 to 1.0 V.

The gravimetric specific capacitance (*C_g_*) was calculated from the CV or galvanostatic charge/discharge (GCD) curves according to Equation (1) [29,30,31,32]:(1)$$C_{g}=\frac{2}{mv∆V}\int IdV\;\mathrm{or}\;C_{g}=\frac{2I}{m(dV/dt)}$$
where *m* is the mass of the active electrode, *ν* is the scan rate, Δ*V* is the potential window, *I* is the discharge current, and *t* is the discharge time.

The gravimetric energy and power densities (*E_g_* and *P_g_*, respectively) of the samples were calculated according to Equation (2) [29,31,33]:(2)$$E_{g}=\frac{1}{8}C_{g}∆V^{2}/3600,\;P_{g}=I∆V/2m$$

## 3. Results and Discussion

Figure 1a provides an overview of the steps involved in creating highly porous carbon structures, starting with the carbonization of chitosan powder for 2 h at 400 °C under an Ar gas atmosphere. *Act*-chitosan is activated with KOH at a 1:4 mass ratio mixture while HNO_3_-*Act*-chitosan is prepared by an acid soak treatment of the carbonized material with HNO_3_, followed by mixing with KOH at a mass ratio of 1:4 and activation for 1 h at 800 °C. The microstructural properties of the carbonized sample (*Carb*-chitosan), *Act*-chitosan, and HNO_3_-*Act*-chitosan were evaluated using SEM; the results are presented in Figure 1b–d. *Carb*-chitosan had a cylindrical shape, with particles ~200 μm in length and a smooth surface without visible pores (Figure 1b). HNO_3_-*Act*-chitosan had a hierarchical porous structure with a carbon skeleton morphology and particles 100–200 μm in length. This structure, formed by carbon walls, consisted of randomly interconnected macropores, as depicted in Figure 1c. *Act*-chitosan consisted of smaller carbon-based platelets ~100 μm in size, but without evident macro-pores (Figure 1d). These findings demonstrate that the morphological properties of the samples can be readily altered by HNO_3_ treatment prior to KOH activation. The textural properties of the samples, including the SSA, pore volume, and PSD, are presented in Figure 2.

To further investigate the textural properties of *Carb*-chitosan carbon and its activated samples, a low-temperature nitrogen adsorption/desorption isotherm analysis was performed at 77 K. As shown in Figure 2a, while type III curves were obtained for *Carb*-chitosan, the two activated chitosan samples were characterized by type I curves. That is, the nitrogen adsorption isotherms showed a high adsorbed amount within a low relative pressure region (P/P_0_ < 0.01), and then a broad knee with a horizontal plateau in the measured relative pressure range, but no clear hysteresis pattern. These results indicated that the KOH-activated materials contained a high abundance of micropores (with some mesopores), which plays a beneficial role in electrochemical reactions. In addition, the isotherm shape can be seen to substantially depend on the pretreatment conditions. Compared to *Act*-chitosan, HNO_3_-*Act*-chitosan had an overall lower amount of adsorption and a gentler slope at a relative pressure (P/P_0_) of 0.05–0.4. The difference can be partially explained by changes in the pore structure, as the carbon skeleton is weakened by HNO_3_ etching, which reduces the surface area and pore volume but increases the proportion of mesopore regions. The PSDs of the three sample types, calculated using the 2D-NLDFT method, are shown in Figure 2b. *Carb*-chitosan had a large number of mesopores (>2 nm) without significant micropores whereas the PSD curves of both activated samples were consistent with predominantly developed ultra-micropores (pore width < 1.0 nm), micropores (pore width < 2 nm), and mesopores (>2 nm). Specifically, HNO_3_-*Act*-chitosan and *Act*-chitosan were dominated by ultra-micropores of 0.89 and 0.91 nm and micropores of 1.98 and 1.93 nm, respectively, with few mesopore regions. The textural property values (SSA, pore volume, and pore width) increased significantly after KOH and HNO_3_-KOH activation (Table 1). The effect of HNO_3_ treatment on pore formation in the carbonized sample was not significant compared to the effect of KOH activation. The calculated BET surface area and pore volume of *Act*-chitosan were 4128 (2731 for 2D-NLDFT) m^2^/g and 1.87 cm^3^/g, which are 1.60 (1.56 for 2D-NLDFT) and 1.32 times larger, respectively, than the values determined for HNO_3_-*Act*-chitosan. In addition, the ratio of the micropore volume to the total pore volume (V_0.2 nm_/V_tot_) for *Act*-chitosan was 71.3%, which was 1.45 times larger than that of HNO_3_-*Act*-chitosan, implying that pretreatment with an aqueous solution of HNO_3_ did not effectively enhance the development of pore networks in the medium. A comparison of the calculated AED curves for *Carb*-chitosan, HNO_3_-*Act*-chitosan, and *Act*-chitosan (Figure 2c) showed that the shape of the curve was strongly dependent on the conditions used in sample preparation. The differences in peak height, peak width, and peak location indicated the presence of different energetic states on the surface of the adsorbent. The unimodal peak of *Carb*-chitosan, located in the range of 3.0–10.1 kJ/mol with a maximum at 6.4 kJ/mol, implied one main energetic state related to the mesoporous region. However, the bimodal peaks of different shapes determined for the two activated samples indicated two different surface energy sites, associated with micropores and mesopores. As shown in Figure 2c, the lower energy peaks related to the mesoporous region were in the same range, 3.0–9.0 kJ/mol, with the same maximum at 6.2 kJ/mol. However, the higher energy peaks, mainly related to the microporous region, were in the range of 9.3–13.5 kJ/mol (HNO_3_-*Act*-chitosan) and 9.3–14.9 kJ/mol (*Act*-chitosan), with the same maximum at 11.4 kJ/mol. The widths of the peaks in the high-energy region suggested greater heterogeneity of the *Act*-chitosan than the HNO_3_-*Act*-chitosan surface, further supporting the higher microporosity ratio of *Act*-chitosan.

The structural properties of the prepared porous samples were further investigated using HR-TEM (Figure 3). A comparison of *Carb*-chitosan (Figure 3a,b), HNO_3_-*Act*-chitosan (Figure 3c,d), and *Act*-chitosan (Figure 3e,f) revealed strikingly distinct morphologies and crystal qualities. Notably, *Act*-chitosan, synthesized solely through the activation process, exhibited a sheet-like microstructure characterized by high-quality graphene-like crystal structures whereas the HNO_3_-*Act*-chitosan displayed the typical morphology of amorphous carbon particles, featuring numerous nanopores with carbon layers arranged in parallel. The *Act*-chitosan samples also had a thinner layering and an interconnected structure, reflecting the sharp lattice fringes of the carbon layers. The selected area electron diffraction pattern (inset in Figure 3f) clearly demonstrated that the carbon atoms formed a hexagonal lattice, indicative of graphene formation and suggesting a reduction in the number of defects in the graphitic carbon structure. Overall, the HR-TEM observations provided compelling evidence for the distinct structural characteristics of the samples and, for *Act*-chitosan, the successful synthesis of graphene-like structures obtained through the activation process. Additionally, the presence of amorphous carbon particles and nanopores in HNO_3_-*Act*-chitosan highlighted the influence of the different preparation methods on the final morphology and crystal quality. Raman spectroscopy is a highly valuable method for discerning the structure, defects, and disordered nature of carbon materials. Appendix A shows that the 1350 cm^−1^ peak (D band) corresponds to the A_1g_ mode of vibration, indicative of disorder or defects, and the 1580 cm^−1^ peak (G band) to the E_2g_ mode of vibration, characteristic of a graphitic structure, as well as stretching of the C–C carbon bond. The intensities of the D-band and G-band reflect the concentration of disordered and graphitized carbon, respectively. For *Act*-chitosan, the D to G intensity ratio was lower than that of HNO_3_-*Act*-chitosan (0.81 and 0.96, respectively). To evaluate the crystalline structure of all samples, an XRD analysis was conducted (Appendix A). The absence of distinct characteristic peaks in the XRD patterns suggested that all samples exhibited the typical structure of disordered carbon material. Two weak broad peaks were detected, at approximately 23 and 43, corresponding to the (002) and (100) reflections of the turbostratic carbon structure, respectively. The broad (100) reflection corresponded to the honeycomb structures formed by sp^2^ hybridized carbons, and the broad (002) reflection to the small domains of graphene sheets with coherent and parallel stacking [34]. Raman spectroscopy and XRD analyses revealed differences in the graphitic structures and concentrations of defects of *Act*-chitosan and HNO_3_-*Act*-chitosan, and provided valuable insights into the structural properties and disordered nature of the carbon materials.

The chemical composition of the samples was thoroughly examined using XPS. The spectra showed three distinct peaks, attributed to carbon (C1s), oxygen (O1s), and nitrogen (N1s), and slightly higher binding energy of sp^3^ hybridized states above 284.5 eV. Upon closer analysis, the high-resolution C1s spectra were deconvoluted, revealing peaks at 284.5, 285.5, and 287.6 eV corresponding to C–H, C–N, and –C=O (encompassing various oxygen-containing functional groups, including –C=O and –COOH [35,36]) species, respectively (Figure 4b). Comparison of the samples showed that *Carb*-chitosan had a lower relative content of sp^2^-hybridized carbon, in contrast to the sharper shape and higher intensity of the *Act*-chitosan spectra. Interestingly, in the *Act*-chitosan sample, while the sp^3^ peak was weak, the strong sp^2^ peak indicated a high degree of graphitization and further evidence of a graphene structure. According to these observations, HNO_3_-*Act*-chitosan and *Act*-chitosan had a similar chemical composition with respect to the nitrogen and oxygen-containing functional groups in their structures, in line with the Raman results demonstrating a higher degree of graphitization. Further analysis of the high-resolution O1s spectra (Figure 4c) revealed two peaks, at 531.7 and 533.3 eV, corresponding to OH and C–O–C, respectively [37]. Although the spectra of HNO_3_-*Act*-chitosan and *Act*-chitosan did not change significantly, a slight decrease in the intensity of the two peaks was observed, as shown in Figure 4c. As shown in Figure 4d, deconvolution of the N1s peaks revealed three components indicative of the presence of N atoms at three different bonding energies at approximately at 398.6 eV, 399.9 eV, and 401.1 eV. These correspond to the pyridine-like peak (–NH_2_, ⅰ), the pyrrol-like peak (O=C=NH–, ⅱ), and nitrogen atoms that are either protonated or oxidized (ⅲ) [17,19,35]. After the acid/KOH activation process of *Carb*-chitosan, a decrease in the total content of nitrogen-related functional groups was observed, and the peaks broadened toward higher binding energies. These results suggest that the acid treatment and KOH activation at elevated temperatures induce changes in the overall composition of nitrogen-related functional groups in the activated chitosan samples.

The electrochemical properties of *Carb*-chitosan, HNO_3_-*Act*-chitosan, and *Act*-chitosan were examined using CV and GCD techniques, with a symmetrical configuration and an aqueous electrolyte solution consisting of 6 M KOH. In CV, a rectangular shape of the curves typically indicates that the energy storage mechanism of an electrode mainly originated from the formation of electrical double layers of electrolyte ions on the electrode surface [31,38,39,40]. As seen in Figure 5a, the CV curves of HNO_3_-*Act*-chitosan and *Act*-chitosan showed almost identical rectangular shapes. The gravimetric capacitance (*C_g_*) values were 139.1 and 183.5 F/g, respectively, at a scan rate of 5 mV/s. Even at a high scan rate of 100 mV/s, the shapes of the CV curves of HNO_3_-*Act*-chitosan and *Act*-chitosan remained rectangular (Figure 5b), with *C_g_* values of 116.7 and 160.5 F/g, respectively. GCD measurements of the specific capacitance as a function of current density yielded results identical to those of the CV measurements and showed the same trend (Appendix A). As seen in Appendix A, the discharging time of *Carb*-chitosan was short, at only 16 s, even at a low current density of 0.15 A/g, whereas the discharge times of HNO_3_-*Act*-chitosan and *Act*-chitosan were 197 s at 0.35 A/g and 178 s at 0.46 A/g, respectively, indicating a higher *C_g_* of activated than carbonized chitosan.

The trends in the rate properties of the samples as a function of a scan rate ranging from 5 mV/s to 500 mV/s are presented in Figure 5c. The far superior rate properties of the ACs, i.e., HNO_3_-*Act*-chitosan and *Act*-chitosan, compared with *Carb*-chitosan could be attributed to the high surface area of the ACs, based on the high degree of microporosity in their pore structure (see the BET analysis results above). This result is consistent with literature reports showing that ACs with a high SSA and high micropore volume as a proportion of the total pore volume have very high specific capacitance [38,39,41]. Unlike the AC samples, *Carb*-chitosan, with its inactivated carbon, had a *C_g_* of only 13.9 F/g, even at a slow scan rate of 5 mV/s, and its CV curve was distorted (Figure 5a,b). These results reflect the fact that inactivated carbon has an insufficient SSA and small pore volume in its internal pore structure, leaving insufficient space for the adsorption of electrolyte ions and thus leading to a low *C_g_* value. During the synthesis of ACs, however, HNO_3_ and/or KOH activation imparts high porosity to the interior of the electrode, resulting in optimized EDLC properties.

Next, EIS was performed to further investigate the effect of the two activation processes on the kinetics of the electrolyte ions for the AC electrodes. The series resistance (*R_s_*) and charge transfer resistance (*R_ct_*) values were extracted from the fitting curve of the Nyquist plot using an equivalent circuit model (Appendix A). *R_s_*, also defined as the internal resistance of the electrode in the electrolyte, is typically taken from the high-frequency region of the Nyquist plot, while *R_ct_*, measured between electrode and electrolyte, is taken from the diameter in the Nyquist plot. As seen in the Nyquist plot of Figure 5d, the *Carb*-chitosan sample had a low *R_s_* value (2.694 Ω), attributed to the introduction of electrical conductivity into the obtained chitosan due to the carbonization process. However, carbonization without the activation process failed to create porosity in the chitosan, resulting in a relatively high *R_ct_* value of about 458.2 Ω. For the two ACs, the *R_s_* and *R_ct_* values of *Act*-chitosan (2.807 Ω and 0.395 Ω, respectively) were lower than those of HNO_3_-*Act*-chitosan (3.667 Ω and 0.438 Ω). This difference between the ACs suggested that the electrolyte ion charge transfer reaction occurred more readily in *Act*-chitosan such that KOH activation is far superior to HNO_3_-KOH activation in terms of improving the electrochemical properties of ACs, i.e., by obtaining a very high porosity and micropore volume.

The cyclic stability of *Carb*-chitosan, *Act*-chitosan, and HNO_3_-*Act*-chitosan was examined by repeating the CV measurements at a scan rate of 100 mV/s for 10,000 cycles, as shown in Figure 5e and Appendix A. *Carb*-chitosan had insufficient *C_g_* values due to its low porosity and showed an initial significant drop in capacitance as the cycle number increased, followed by a tendency toward an increased capacitance as the cycle progressed. This is a well-known property of non-porous carbon electrodes during repeated cycle testing [14,41,42,43]. By contrast, both *Act*-chitosan and HNO_3_-*Act*-chitosan showed stable device behavior and capacitance retention rates of 99.2% and 98.7%, respectively, confirming them as stable electrode materials for EDLCs.

The charge storage mechanism of the fabricated carbons was investigated by analyzing the capacitive contribution to the total specific capacitance, using the log(*i*) versus log(*ν*) plots obtained from the CV experiments, as a function of the scan rate; this method was employed in previous studies [44,45,46]. The CV measurements for each sample were obtained at scan rates of 5, 10, 20, 50, and 100 mV/s. The diffusion-controlled (*i_Diff_*) and capacitive-controlled contributions (*i_Cap_*) of the total specific capacitance were calculated to determine the charge storage and rate-dependent kinetics, as shown in Equation (3):(3)$$i=i_{Diff}+i_{Cap}=a\nu^{b}$$
where *a* and *b* are constants, and *i* is the anodic or cathodic peak current density.

If the value of *b*, obtained from the slope of the log(i) versus log(ν) plot, is close to 1.0, the charge storage is primarily due to surface reactions (i.e., EDLC); if it is close to 0.5, it is due to diffusion-controlled reactions, such as intercalation. The *b*-values for *Act*-chitosan and HNO_3_-*Act*-chitosan were 0.89 and 0.83, respectively, while the *b*-value of *Carb*-chitosan was 0.47. Non-activated *Carb*-chitosan did not have sufficient BET values (13 m^2^/g) and was subject to slow charge storage processes, mainly diffusion-controlled processes. The total capacitance of the AC samples, however, showed a major contribution from the capacitive-control process, with a much stronger EDLC response by *Act*-chitosan than HNO_3_-*Act*-chitosan. This result implied that KOH activation is more suitable than HNO_3_-KOH activation in AC preparation, as AC prepared by the former was much more porous and therefore had a better EDLC response.

Equation (4) was used to quantitatively analyze the capacitive contribution to the total capacitance (capacitive-controlled plus diffusion-controlled currents) (Appendix A):(4)$$i\left(V\right)=k_{1}\nu +k_{2}\nu^{1/2}$$
where *i*(V) is the current at a fixed potential, k_1_ν is the capacitive-controlled current, and k_2_ν^1/2^ is the diffusion-controlled current.

*Carb*-chitosan accounted for only a small fraction of the capacitive-controlled current across the whole range of scan rates due to the low porosity of the electrode, whereas the capacitive contributions of *Act*-chitosan and HNO_3_-*Act*-chitosan to the total capacitance were large. For example, *Act*-chitosan contributed 83% at a scan rate of 5 mV/s and 96% at a scan rate of 100 mV/s. For HNO_3_-*Act*-chitosan, the capacitive contributions were 76% and 93%, respectively. These results implied that *Act*-chitosan and HNO_3_-*Act*-chitosan are not diffusion-controlled processes, but rather surface reactions similar to those that dominate in EDLCs, and the much greater increase in the total capacitance of *Act*-chitosan than HNO_3_-*Act*-chitosan was due to an increase in the EDLC portion of the electrochemical reaction rather than to the diffusion-controlled portion.

The electrochemical performances of *Act*-chitosan and HNO_3_-*Act*-chitosan were compared by calculating their energy and power densities from the *C_g_* data. The results are shown in the Ragone plots in Figure 5f. *Act*-chitosan exhibited a superior gravimetric energy density (*E_g_*) of 6.37 Wh/kg and a high power density of 229.4 W/kg. Surprisingly, at high current densities, *Act*-chitosan maintained an energy density value of 4.49 Wh/kg and a power density as high as 16 kW/kg. These values are comparable to those reported in the literature for other state-of-the-art biomass-derived ACs [39,47,48,49,50,51]. For example, as seen in the Ragone plots (Figure 5f), the energy and power densities of *Act*-chitosan were comparable to those determined in ACs derived from chestnut or chitosan and porous microsphere carbons. However, despite the much improved energy and power densities of biomass-derived carbons, their energy densities are limited compared with the high energy density nanocarbons of N-doped porous carbon and titanium carbide-derived carbon [52]. Therefore, increasing the energy density will require the hybridization of *Act*-chitosan with other capacitive materials, such as metal oxides/nitrides and conductive polymers, or the design of asymmetric device configurations with different types of positive/negative electrodes.

## 4. Conclusions

In this study, a straightforward and efficient activation protocol for producing highly porous ACs derived from chitosan was presented. Following the pyrolysis of the chitosan precursor, both the HNO_3_-soaked sample and the untreated sample underwent subsequent activation through a KOH chemical activation process. This process facilitated the development of porosity, resulting in a substantial specific surface area, a high pore volume, and the formation of diverse particle shapes. The electrochemical performance of these materials in a 6 M KOH aqueous electrolyte was then evaluated. An N_2_ adsorption/desorption analysis showed that the BET SSA and pore volume of *Act*-chitosan reached 4128 m^2^/g and 1.87 cm^3^/g, respectively. To our knowledge, these are among the highest reported values for ACs. Furthermore, we conducted an evaluation of the connection between the material’s morphological and structural properties and its capacitance. The *Act*-chitosan-based electrode, distinguished by its larger specific surface area and increased pore volume with pores below 2 nm, demonstrated a notable enhancement in EDLC performance. The optimized activated chitosan exhibited outstanding capacitance performance, achieving approximately 183.5 F/g at a scan rate of 5 mV/s and a superior gravimetric energy density (*E_g_*) of 6.37 Wh/kg with a high power density of 229.4 W/kg. Even after enduring 10,000 cycles, it consistently maintained stable capacitance. Moreover, from calculations about the portion of capacitive contribution in the total capacitance, the *Act*-chitosan-based electrode revealed a large portion in the total capacitance in terms of 83% at a scan rate of 5 mV/s and 96% even at a fast scan rate of 100 mV/s, respectively. The exceptional electrochemical performance of chitosan-induced ACs was attributed to the well-developed conductive sites present within the optimized pores. This unique attribute enhanced ion migration and charge storage capabilities, ultimately leading to the excellent capacitance performance of *Act*-chitosan.

## Figures and Tables

**Figure 1 nanomaterials-13-02961-f001:**
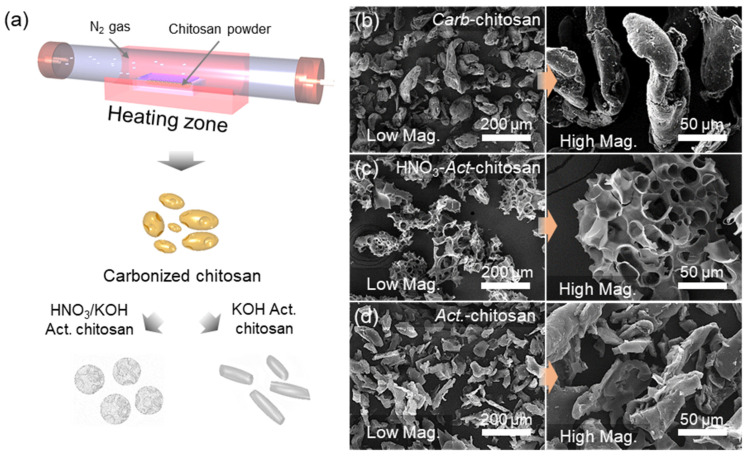
Schematic illustration and characterization of carbonized chitosan (*Carb*-chitosan) and the activated samples (HNO_3_-*Act*-chitosan and *Act*-chitosan). (**a**) Experimental steps. SEM images (low magnification [**left**] and high magnification [**right**]). (**b**) *Carb*-chitosan fabricated at 400 °C. (**c**) HNO_3_-*Act*-chitosan activated at 800 °C after HNO_3_ treatment. (**d**) *Act*-chitosan activated at 800 °C.

**Figure 2 nanomaterials-13-02961-f002:**
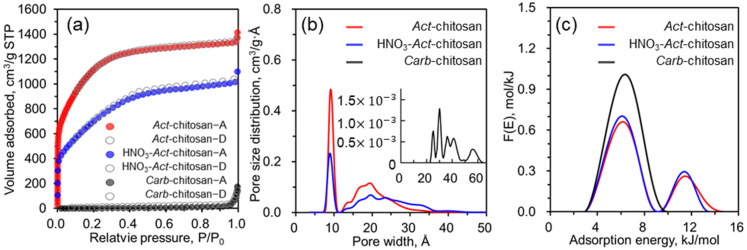
(**a**) N_2_ adsorption/desorption isotherm of *Carb*-chitosan, HNO_3_-*Act*-chitosan, and *Act*-chitosan measured at 77 K. (**b**) Pore size distribution calculated using the 2D-NLDFT equation. (**c**) Adsorption energy distribution.

**Figure 3 nanomaterials-13-02961-f003:**
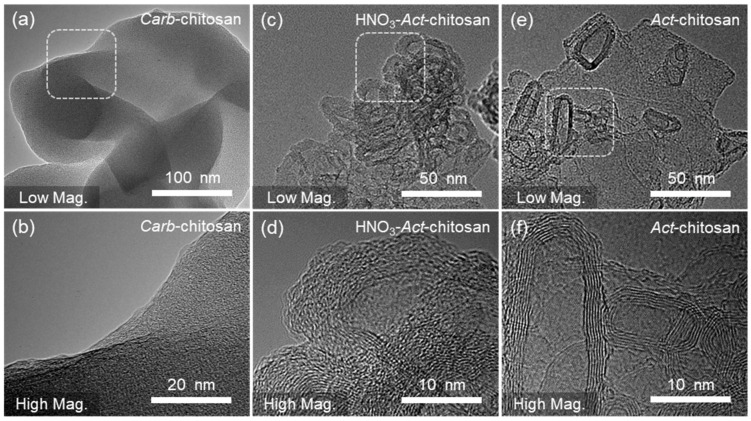
TEM images of *Carb*-chitosan, HNO_3_-*Act*-chitosan, and *Act*-chitosan. (**a**,**b**) *Carb*-chitosan (**c**,**d**) HNO_3_-*Act*-chitosan. (**e**,**f**) *Act*-chitosan. Low magnification (**top**) and high magnification (**bottom**).

**Figure 4 nanomaterials-13-02961-f004:**
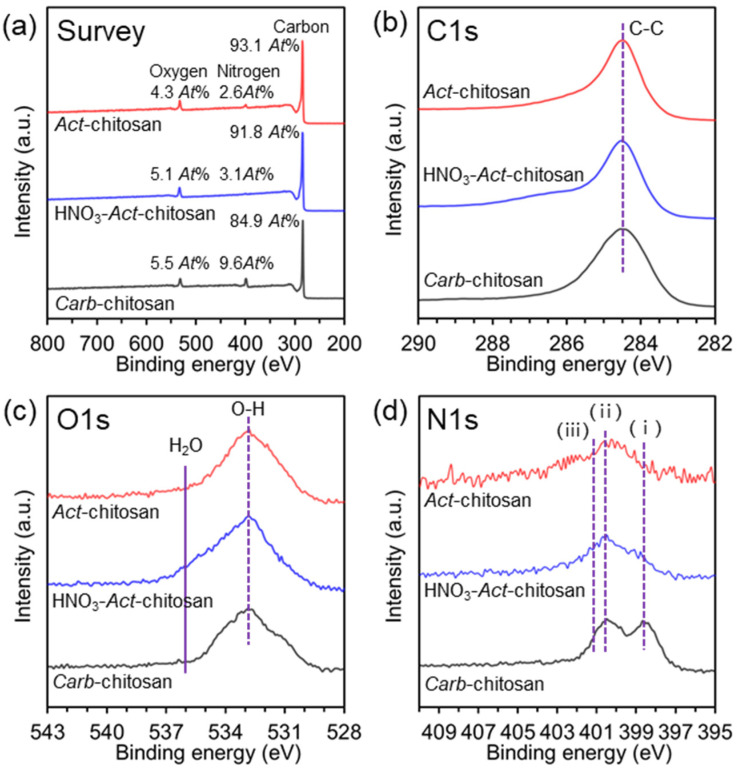
Chemical composition of *Carb*-chitosan, HNO_3_-*Act*-chitosan, and *Act*-chitosan as measured by XPS. (**a**) Survey. (**b**) C1s narrow-scan spectra. (**c**) O1s narrow-scan spectra. (**d**) N1s narrow-scan spectra. (i) Pyridine-like peak (–NH_2_), (ii) Pyrrol-like peak (O=C=NH–), (iii) nitrogen atoms that are either protonated or oxidized.

**Figure 5 nanomaterials-13-02961-f005:**
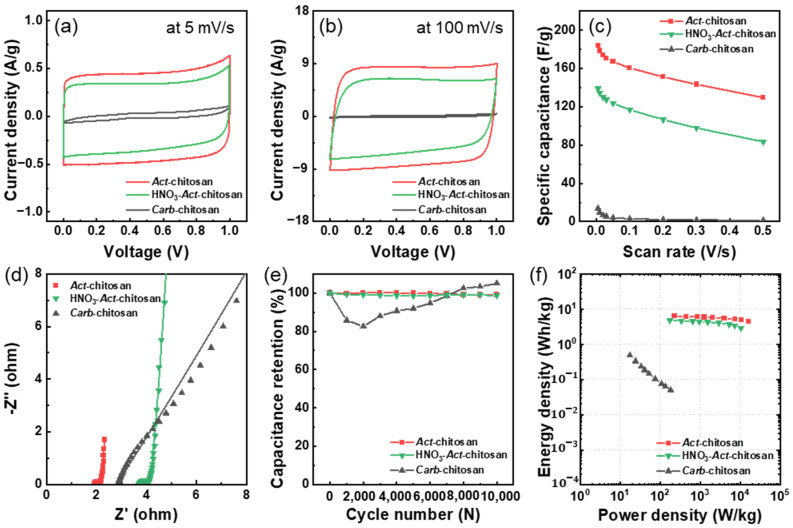
Cyclic voltammetry (CV) curves of *Carb*-chitosan, HNO_3_-*Act*-chitosan, and *Act*-chitosan at scan rates of (**a**) 5 mV/s and (**b**) 100 mV/s. (**c**) Rate performance of the samples at scan rates of 5–500 mV/s. (**d**) Nyquist plots of the samples. (**e**) Capacitance retention of the electrodes during 10,000 cycles. (**f**) Ragone plots of the specific energy density and power density of the supercapacitor for each electrode.

**Table 1 nanomaterials-13-02961-t001:** Textural parameters of *Carb*-chitosan, HNO_3_-*Act*-chitosan, and *Act*-chitosan.

Sample	Surface Area (m^2^/g)		Pore Volume (cm^3^/g)		Pore Width (nm)
	S_BET_	S_2D-NLDFT_	V_2D-NLDFT, <2 nm_	V_2D-NLDFT, >2 nm_	W_2D-NLDFT_
*Carb*-chitosan	13	8	0	0.013	17
HNO_3_-*Act*. chitosan	2584	1765	0.701	0.718	0.89/1.98
*Act*-chitosan	4128	2731	1.338	0.539	0.91/1.93

## Data Availability

Data are contained within the article.

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
