# Peer review of "Hierarchically Porous Carbon Networks Derived from Chitosan for High-Performance Electrochemical Double-Layer Capacitors"

_nanomaterials, 2023, doi:10.3390/nano13222961_

Round 1
Reviewer 1 Report
Comments and Suggestions for Authors
In this study, the Authors present novel macro/micro-porous carbon structures synthesized from a chitosan precursor using an acid/KOH activation process and investigate the relationship between the material's textural characteristics and its capacitance in an electric double-layer capacitor (EDLC). However, some general points and some additional remarks in the text need to be addressed before publication:
1. The Introduction needs improvement. Given the presented data and existing literature, this work's presentation should highlight why this work is necessary and what new insights on the presented problems are expected.
2. Conclusion
The summary part needs improvement. The summary of the research conducted is trivial and repeats the data contained in the abstract. The purposefulness of the study and its potential application in practice should be emphasized more strongly.
This manuscript will be interesting for all scientists involved in such studies, and I suggest publication after the minor revision of the paper.
Author Response
#Reviewer 1
In this study, the Authors present novel macro/micro-porous carbon structures synthesized from a chitosan precursor using an acid/KOH activation process and investigate the relationship between the material's textural characteristics and its capacitance in an electric double-layer capacitor (EDLC). However, some general points and some additional remarks in the text need to be addressed before publication:
- The Introduction needs improvement. Given the presented data and existing literature, this work's presentation should highlight why this work is necessary and what new insights on the presented problems are expected.
[Response]
We are grateful to the reviewer for evaluating our manuscript. We also doubly thank you for your comments of our manuscript. According to your comment, we added additional statement for highlight in our work compared to previous works in the revised manuscript.
Revised statement in introduction of the revised manuscript
Moreover, the presence of a limited number of active sites, attributed to randomly developed pores, irregular particle shapes, and low production yields, presents persistent challenges. Therefore, it is essential to establish a scalable synthesis method for active materials with uniformly distributed nanopores measuring less than 2nm. This synthesis approach is pivotal for achieving enhanced performance metrics, including increased capacitance, retention of capacitance at higher scan rates, and consistent capacitance throughout extended cycles in energy storage systems.
- Conclusion
The summary part needs improvement. The summary of the research conducted is trivial and repeats the data contained in the abstract. The purposefulness of the study and its potential application in practice should be emphasized more strongly.
This manuscript will be interesting for all scientists involved in such studies, and I suggest publication after the minor revision of the paper.
[Response]
We appreciate the invaluable comment of the reviewer. According to reviewer comments, we revised conclusion in the revised manuscript.
Revised statement in conclusion of the revised manuscript
In this study, a straightforward and efficient activation protocol for producing highly porous ACs derived from chitosan was presented. Following the pyrolysis of the chitosan precursor, both the HNO3-soaked sample and the untreated sample underwent subsequent activation through a KOH chemical activation process. This process facilitated the development of porosity, resulting in a substantial specific surface area, a high pore volume, and the formation of diverse particle shapes. The electrochemical performance of these materials in a 6 M KOH aqueous electrolyte was then evaluated. An N2 adsorption/desorption analysis showed that the BET SSA and pore volume of Act-chitosan reached 4128 m²/g and 1.87 cm3/g, respectively. To our knowledge, these are among the highest reported values for ACs. Furthermore, we conducted an evaluation of the connection between the material's morphological and structural properties and its capacitance. The Act-chitosan-based electrode, distinguished by its larger specific surface area and increased pore volume with pores below 2 nm, demonstrated a notable enhancement in EDLC performance. The optimized activated chitosan exhibited outstanding capacitance performance, achieving approximately 183.5 F/g at a scan rate of 5 mV/s and a superior gravimetric energy density (Eg) of 6.37 Wh/kg with a high power density of 229.4 W/kg. Even after enduring 10,000 cycles, it consistently maintained stable capacitance. Moreover, from calculation about the portion of capacitive contribution in the total capacitance, Act-chitosan-based electrode revealed a large portion in the total capacitance in terms of 83% at a scan rate of 5 mV/s and 96% even at a fast scan rate of 100 mV/s, respectively. The exceptional electrochemical performance of chitosan-induced ACs was attributed to the well-developed conductive sites present within the optimized pores. This unique attribute enhanced ion migration and charge storage capabilities, ultimately leading to the excellent capacitance performance of Act-chitosan.

Reviewer 2 Report
Comments and Suggestions for Authors
I am impressed by the research and the description of it that the authors have provided. The goals are clearly stated, as is the importance of achieving them. I am convinced that the researchers have made significant progress toward the fabrication of energy storage systems derived from biomass.
Details associated with the pre-treatment of the commercially available chitosan and subsequent activation protocols is described in section 2.1 of the manuscript. I believe that the authors have provided sufficient detail so that other researchers should be able to reproduce the results. The names Carb-chitosan, HNO3-Act-chitosan, and Act-chitosan highlight the way in which chitosan was treated in order to generate electrode slurries. For example, Act-chitosan refers to chitosan that was activated by heating to 800 °C, while HNO3-Act-chitosan refers to chitosan that was pretreated with HNO3 and subsequently further activated with KOH. Meanwhile Carb-chitosan refers to chitosan fabricated at 400 ℃. Each of these fabrications lead to different morphologies and differing properties, as summarized in Table 1 and Figures 1-5 [Table 1 (textural parameters), SEM images (Fig. 1), isotherms (Fig. 2), TEM (Fig. 3), XPS (Fig. 4), and in Fig. 5, CV, capacitance, Nyquist and Ragone plots]. The outcomes are intriguing, with Act-chitosan exhibiting the most impressive properties. Its large specific surface area coupled with a high pore volume, for example, contributed to its excellent capacitance performance even after repeated cycles (10,000). This outcome is integral to achieving one of the overarching goals of the research viz., the ability to use biomass derived material as a source of high capacitance electrochemical energy storage systems (e.g., electrochemical double layer capacitors).
Author Response
#Reviewer 2
I am impressed by the research and the description of it that the authors have provided. The goals are clearly stated, as is the importance of achieving them. I am convinced that the researchers have made significant progress toward the fabrication of energy storage systems derived from biomass.
Details associated with the pre-treatment of the commercially available chitosan and subsequent activation protocols is described in section 2.1 of the manuscript. I believe that the authors have provided sufficient detail so that other researchers should be able to reproduce the results. The names Carb-chitosan, HNO3-Act-chitosan, and Act-chitosan highlight the way in which chitosan was treated in order to generate electrode slurries. For example, Act-chitosan refers to chitosan that was activated by heating to 800 °C, while HNO3-Act-chitosan refers to chitosan that was pretreated with HNO3 and subsequently further activated with KOH. Meanwhile Carb-chitosan refers to chitosan fabricated at 400 ℃. Each of these fabrications lead to different morphologies and differing properties, as summarized in Table 1 and Figures 1-5 [Table 1 (textural parameters), SEM images (Fig. 1), isotherms (Fig. 2), TEM (Fig. 3), XPS (Fig. 4), and in Fig. 5, CV, capacitance, Nyquist and Ragone plots]. The outcomes are intriguing, with Act-chitosan exhibiting the most impressive properties. Its large specific surface area coupled with a high pore volume, for example, contributed to its excellent capacitance performance even after repeated cycles (10,000). This outcome is integral to achieving one of the overarching goals of the research viz., the ability to use biomass derived material as a source of high capacitance electrochemical energy storage systems (e.g., electrochemical double layer capacitors).
[Response]
We appreciate the reviewer for assessing our manuscript. Furthermore, our manuscript has undergone thorough English language checking and correction through the Textcheck service, and the Textcheck certificate is provided as an attachment.

Reviewer 3 Report
Comments and Suggestions for Authors
This manuscript describes the fabrication of chitosan-based porous carbon materials and application to high performance electrochemical capacitor. The obtained materials were fully characterized by various methods and the obtained results were discussed reasonably. Besides, their capacitances were sufficiently high compared to previously reported ones. As this manuscript includes the valuable information, I recommend this article for the publication in Nanomaterials. However, I also recommend the minor revision to improve the quality of this manuscript. My comments are below.
1. I fully understood the treatment of chitosan with KOH or the HNO3 and KOH afforded the nanoporous materials. However, why was such a porous structure formed? Can the authors discuss the difference in terms of their chemical structures?
2. Page 7, line 6. Pyridine nitrogen should be NH not NH2. As similar, pyridinium nitrogen should be NH2+ not NH3+.
3. Figure 4. The shape of N1s peak of Carb-chitosan is very different from that of HNO3-Act-chitosan and Act-chitosan. Please add some comments about the change of chemical structures based on this result if possible.
4. Figure 4d. The authors assigned C–O peak at 401 eV. Is it correct?
5. Figure 5. The legend “Carbonized Chitosan” should be changed “Carb-chitosan” as shown in the main sentence.
6. Typos. P3. Act-chitosan’s → Act-chitosan’s; P6. sp2, sp3 → sp2, sp3; P9. Act. Chitosan → Act-chitosan.

Minor editing of English language required.
Author Response
#Reviewer 3
This manuscript describes the fabrication of chitosan-based porous carbon materials and application to high performance electrochemical capacitor. The obtained materials were fully characterized by various methods and the obtained results were discussed reasonably. Besides, their capacitances were sufficiently high compared to previously reported ones. As this manuscript includes the valuable information, I recommend this article for the publication in Nanomaterials. However, I also recommend the minor revision to improve the quality of this manuscript. My comments are below.
- I fully understood the treatment of chitosan with KOH or the HNO3 and KOH afforded the nanoporous materials. However, why was such a porous structure formed? Can the authors discuss the difference in terms of their chemical structures?
[Response]
We appreciate the reviewer’s comments and suggestions. Generally, Nitric acid (HNO3) treatment provides a convenient and direct approach for introducing surface functional groups onto carbon materials, encompassing carboxylic, phenolic, quinone, nitro, and peroxide groups. These functional groups possess the potential to induce significant alterations in the morphological, textural, and surface chemistry properties of carbons. This transformation is attributed to chemical reactions, gas evolution, and etching/activation processes, particularly under high-temperature activation conditions [Journal of Power Sources 236 (2013) 285-292, Journal Power Sources, 195 (2010), 7880-7903, Journal of Energy Storage 72 (2023) 108506]. In this context, our findings suggest that the KOH activation of carbonized chitosan immersed in an HNO3 solution may result in more pronounced morphological and chemical changes compared to those observed in Act-chitosan. Notably, Act-chitosan observed notable development of porosity such as a large specific surface area and high pore volume, characterized by palate-shaped particles.
- Page 7, line 6. Pyridine nitrogen should be NH not NH2. As similar, pyridinium nitrogen should be NH2+ not NH3+.
[Response]
Thank you for your invaluable comments. Typically, the N1s spectra can be deconvoluted into three distinct peaks with binding energies approximately at 398.6 eV, 399.9 eV, and 401.1 eV. These correspond to the pyridine-like peak (-NH2), the pyrrol-like peak (O=C=NH-), and nitrogen atoms that are either protonated or oxidized [Nano Energy 15 (2015) 9-23, Sustain. Energy Fuels 3 (2019) 1215-1224, Nano Lett. 14 (2014) 4306-4313, Water Res. 38 (2004) 2424–2432, ACS Appl. Mater. Interfaces 2 (2010) 1707–1713, RSC Adv. 5 (2015) 62778-62787, Sci. Rep. 8 (2018) 15397]. Accordingly, we revised the related-statement in the revised manuscript.
Revised statement in the revised manuscript
As shown in Figure 4d, deconvolution of the N1s peaks revealed three components indicative of the presence of N atoms at three different bonding energies at approximately at 398.6 eV, 399.9 eV, and 401.1 eV. These correspond to the pyridine-like peak (-NH2), the pyrrol-like peak (O=C=NH-), and nitrogen atoms that are either protonated or oxidized [17,19,35].
- Figure 4. The shape of N1s peak of Carb-chitosan is very different from that of HNO3-Act-chitosan and Act-chitosan. Please add some comments about the change of chemical structures based on this result if possible.
[Response]
Thank you for comments. In our work, after the acid/KOH activation process of Carb-chitosan, a decrease in the total content of nitrogen-related functional groups was observed, and the peaks broadened toward higher binding energies. These results suggest that the acid treatment and KOH activation at elevated temperatures induce changes in the overall composition of nitrogen-related functional groups in the samples. Accordingly, we added additional statement about change of chemical composition after acid/KOH activation in the revised manuscript.
Added statement in the revised manuscript
After the acid/KOH activation process of Carb-chitosan, a decrease in the total content of nitrogen-related functional groups was observed, and the peaks broadened toward higher binding energies. These results suggest that the acid treatment and KOH activation at elevated temperatures induce changes in the overall composition of nitrogen-related functional groups in the activated chitosan samples.
- Figure 4d. The authors assigned C-O peak at 401 eV. Is it correct?
[Response]
Thank you for invaluable comment. The typo regarding the C-N peak was corrected to the protonated or oxidized nitrogen atoms in the revised manuscript.
- Figure 5. The legend “Carbonized Chitosan” should be changed “Carb-chitosan” as shown in the main sentence.
[Response]
Thank you for invaluable comment. According to reviewer comment, the legend about carbonized chitosan in Figure 5 was revised to Carb-chitosan.
- Typos. P3. Act-chitosan’s → Act-chitosan’s; P6. sp2, sp3 → sp2, sp3; P9. Act. Chitosan → Act-chitosan.
[Response]
Thank you for invaluable comment. According to reviewer comment, the typos were corrected in the revised manuscript.

Reviewer 4 Report
Comments and Suggestions for Authors
The manuscript by K. H. Park and coworkers describes the preparation of carbon materials for supercapacitor applications from chitosan by thermolysis. The materials metrics are reasonable (including ultrahigh surface area based on their activation method), and the capacitance properties are sufficient. The authors have used their results to assess the availability and activity of electrochemical sites making this study somewhat more informative than other similar studies so that I can recommend acceptance of this work.
Comments on the Quality of English LanguageEnglish language needs minor brushing up throughout.
Author Response
#Reviewer 4
The manuscript by K. H. Park and coworkers describes the preparation of carbon materials for supercapacitor applications from chitosan by thermolysis. The materials metrics are reasonable (including ultrahigh surface area based on their activation method), and the capacitance properties are sufficient. The authors have used their results to assess the availability and activity of electrochemical sites making this study somewhat more informative than other similar studies so that I can recommend acceptance of this work.
[Response]
We are thankful to the reviewer for evaluating our manuscript. Additionally, our manuscript has been meticulously reviewed and corrected for English language issues using the Textcheck service. The Textcheck certificate confirming this process is included as an attachment.
